# Long-RVOS: A Comprehensive Benchmark for Long-term Referring Video Object Segmentation

## Abstract

Referring video object segmentation (RVOS) aims to identify, track and segment the objects in a video based on language descriptions, which has received great attention in recent years. However, existing datasets remain focus on short video clips within several seconds, with salient objects visible in most frames. To advance the task towards more practical scenarios, we introduce **Long-RVOS**, a large-scale benchmark for long-term referring video object segmentation. Long-RVOS contains 2,000+ videos of an average duration exceeding 60 seconds, covering a variety of objects that undergo occlusion, disappearance-reappearance and shot changing. The objects are manually annotated with three different types of descriptions to individually evaluate the understanding of static attributes, motion patterns and spatiotemporal relationships. Moreover, unlike previous benchmarks that rely solely on the per-frame spatial evaluation, we introduce two new metrics to assess the temporal and spatiotemporal consistency. We benchmark 6 state-of-the-art methods on Long-RVOS. The results show that current approaches struggle severely with the long-video challenges. To address this, we further propose ReferMo, a promising baseline method that integrates motion information to expand the temporal receptive field, and employs a local-to-global architecture to capture both short-term dynamics and long-term dependencies. Despite simplicity, ReferMo achieves significant improvements over current methods in long-term scenarios. We hope that Long-RVOS and our baseline can drive future RVOS research towards tackling more realistic and long-form videos. Our dataset and code will be released.

## 1 Introduction

Referring Video Object Segmentation (RVOS) [2, 7, 44] is an emerging task that aims to identify, track and segment the object in the video based on a natural language description. Unlike traditional semi-supervised VOS models that require first-frame masks as the object prompt, RVOS models rely solely on text descriptions to segment the target. Considering its potential applications like video editing, growing efforts have been devoted to this field [7, 15, 25, 30, 33]. Recently, the advent of multi-modal large language models [17, 27, 51] and segment anything models [21, 35] has further accelerated this progress [1, 47, 50, 54].

Despite these advances, current RVOS datasets [7, 11, 20, 36] remain limited to short video clips lasting only a few

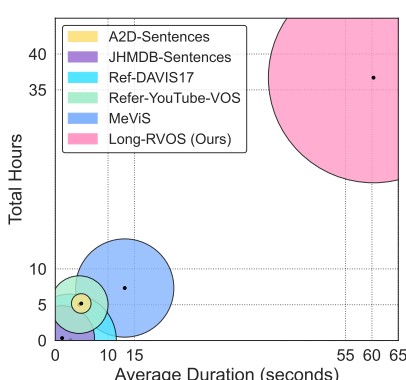

Figure 1: Duration comparison of current RVOS datasets. The circle size indicates the number of frames.

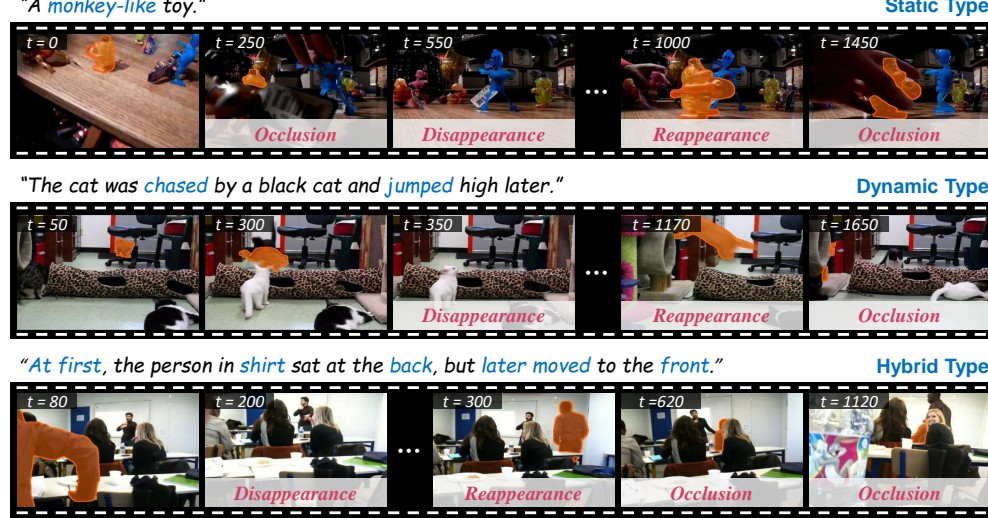

Figure 2: Examples from Long-RVOS dataset, with frame indices displayed in the upper left, and selected objects masked in orange ■. Long-RVOS contains extensive long-term videos, where the objects always undergo occlusion, disappearance-reappearance and shot changing. In addition, the objects are annotated with three different types descriptions: *static*, *dynamic* and *hybrid*.

seconds, with target objects clearly visible in most frames. For state-of-the-art (SOTA) methods, in order to capture the target object effectively, it is inevitable to integrate the text and spatiotemporal information throughout the video. However, when the video becomes longer, the number of distractors also increase accordingly, making it more challenging to perform sufficient spatiotemporal reasoning and capture the key information. Especially in RVOS, many text descriptions (e.g., "the cat jumps down") only refer to a brief fragment in the video. In addition, due to the GPU memory limitation, existing methods typically sample 4∼8 frames per video for training, but use all the frames during inference. As the video length increases, the gap between training and inference phases may become more pronounced. Despite these concerns, due to the lack of a long-term RVOS dataset, the exact challenges posed by longer videos remain unclear.

Another concern lies in the evaluation metrics. Existing RVOS benchmarks [7, 11, 20, 36] typically evaluate performance by simply averaging the frame-wise segmentation metrics (e.g., $\mathcal{J}\&\mathcal{F}$). However, in real-world videos, the target objects do not appear in every frame, due to occlusion and constrained camera views. Therefore, a robust RVOS model should exhibit a sound temporal consistency. This means it should not only accurately segment the target when it is present, but also be able to predict its absence by outputting an empty mask. However, this capability of temporal consistency can not be adequately reflected by current metrics.

To address these gaps, this work proposes **Long-RVOS**, a large-scale benchmark for long-term video object segmentation. Long-RVOS is the first minute-level dataset in RVOS field, designed to tackle various realistic long-video challenges such as frequent occlusion, disappearance-reappearance and shot changing, as shown in Figure 1 and Figure 2. Additionally, we introduce two new metrics for better evaluation of temporal consistency: tIoU, which measures the temporal overlap between predicted and ground-truth mask sequences; and vIoU, which further measures the spatiotemporal volume overlap between them. We benchmark 6 state-of-the-art (SOTA) methods on Long-RVOS. The results demonstrate that while notable progress has been achieved in existing short-term benchmarks, these SOTA models still significantly struggle in realistic long-term scenarios, in both frame-level segmentation and video-level temporal consistency.

To tackle the challenges posed by Long-RVOS, we present a baseline method **ReferMo**, which integrates additional motion frames to expand the temporal receptive field during training, and employs a local-to-global architecture to perceive both static attributes, short-term dynamics and long-term dependencies. Specifically, ReferMo decomposes each video into a sequence of clips, each consisting of a high-resolution keyframe and multiple low-resolution motion frames. Then, it perceives the static appearance and short-term motion within local video clip, and captures the global target in long-term context via inter-clip interactions. In this way, the temporal receptive field is

Table 1: Statistical overview of representative RVOS datasets. Long-RVOS features the longest video duration and the most diverse object classes. Besides, Long-RVOS offers explicit text description types for finer-grained evaluation.

| Dataset | Year | Videos | Mean duration | Total duration | Mean frames | Masks | Objects | Object classes | Text | Text type |
|---------|------|--------|---------------|----------------|-------------|-------|---------|----------------|------|-----------|
| A2D-Sentences [11] | 2018 | 3,782 | 4.9s | 5.2h | 3.2 | 58k | 4,825 | 6 | 6,656 | ✗ |
| JHMDB-Sentences [11] | 2018 | 928 | 1.3s | 0.3h | 34.3 | 32k | 928 | 1 | 928 | ✗ |
| Ref-DAVIS17 [20] | 2018 | 90 | 2.9s | 0.1h | 69.0 | 14k | 205 | 78 | 1,544 | ✗ |
| Refer-YouTube-VOS [36] | 2020 | **3,978** | 4.5s | 5.0h | 27.2 | 131k | 7,451 | 94 | 15,009 | ✗ |
| MeViS [7] | 2023 | 2,006 | 13.2s | 7.3h | 79.0 | 443k | **8,171** | 36 | **28,570** | ✗ |
| **Long-RVOS** (ours) | 2025 | 2,193 | **60.3s** | **36.7h** | **361.7** | **2.1M** | 6,703 | **163** | 24,689 | ✓ |

expanded from multiple frames to multiple clips, but the training cost does not increase significantly. Despite simplicity, ReferMo achieves significant improvements over existing RVOS approaches, serving a promising baseline for long-term referring video object segmentation.

Our contributions are summarized as follows: (i) We build Long-RVOS, the first large-scale benchmark for long-term RVOS. In Long-RVOS, we provide explicit description types and introduce new metrics to enable more comprehensive evaluation. (ii) We benchmark 6 state-of-the-art RVOS approaches on Long-RVOS, and propose a promising baseline ReferMo to address the challenges in long-video scenarios. These contributions establish a foundation for developing more robust RVOS models to handle the realistic long-term videos.

## 2  Related Works

**RVOS Benchmarks.** Given an object description, RVOS aims to identify, tracking and segment the referring object throughout the video. This task was initially introduced by Gavrilyuk et al. [11] and Khoreva et al. [20] in 2018, and has gradually become a popular topic in vision-language understanding. Gavrilyuk et al. [11] built A2D-Sentences and JHMDB-Sentences datasets, which focus on distinguishing different actors in a video through the descriptions about appearance and actions. Khoreva et al. [20] built Ref-DAVIS17 [20], which covers more diverse object types. Later, Ref-Youtube-VOS [36] was developed to further expand the benchmark scale in this field. Recently, MeViS [7] was proposed to highlight the importance of motion understanding in RVOS task. Despite the efforts, these benchmarks remain limited to short video clips lasting only a few seconds, with target objects clearly visible in most frames. Besides, they also lack sufficient evaluation mechanisms to consider the models' specific capabilities in various aspects.

**RVOS Approaches.** Recent RVOS approaches are mainly based on Transformer-based end-to-end architecture, represented by MTTR [2] and ReferFormer [44]. For an effective and consistent object identification across the frames, follow-up works [14, 15, 30, 39] focus on integrating more object-level temporal information. ReferDINO [25] further improves the object-level visual-language understanding by inheriting the object grounding capability of GroundingDINO [28]. Meanwhile, the recent emergence of segment anything models, i.e., SAM [21] and SAM2 [35], provides unique opportunity for downstream segmentation tasks. Some frontier studies [1, 5, 26, 47, 50] explore to incorporate SAM and SAM2 into RVOS approaches, achieving significant improvements on existing benchmarks. For example, VideoLISA [1] incorporates large language models with SAM for reasoning video segmentation. SAMWISE [5] integrate text prompts into SAM2 by inserting trainable adapters. While these models achieve great progress in current short-video benchmarks, their abilities and robustness in handling real-world long videos is still unclear.

**Long-term Video Understanding.** Real-world videos are always long, untrimmed, and involves multiple events. To promote research into long-term video understanding, many large-scale benchmarks [3, 10, 31, 43] have been constructed. However, these benchmarks are mainly constructed for video question answering and temporal action localization, containing only sparse annotations such as timestamps, action labels and captions. To support object-level long-term understanding, some datasets including VidOR [37] and LaSOT [9] also provide dense annotations of bounding boxes. However, long-video datasets with pixel-level dense annotations are still very scarce. Recently, LVOS [16] is built for long-term video object segmentation. However, it is limited in scale and lacks text annotation. In this work, we build Long-RVOS, the first large-scale benchmark for long-term video object segmentation, providing both pixel-wise annotations and diverse object descriptions.

## 3 Long-RVOS: A Comprehensive Benchmark for Long-term RVOS

### 3.1 Video Collection

Previous RVOS datasets [7, 11, 20, 36] were typically constructed by providing text annotations on their corresponding VOS datasets (e.g., DAVIS17 [34], YouTube-VOS-2019 [46] and MOSE [8]). However, the existing long-term VOS datasets like LVOS [16] are limited in scale (containing only 720 videos), and most videos feature only one object target. Therefore, in order to establish a large-scale and diverse RVOS benchmark, we bypass the existing VOS datasets and turn to integrate multi-source long video datasets. Specifically, we build Long-RVOS based on three long-video datasets: TAO [6], VidOR [37], and Ego-Exo4D [12]. Moreover, TAO is a federated dataset combining multiple sources like Charades [38], LaSOT [9], ArgoVerse [4], AVA [13], YFCC100M [41], BDD-100K [49], and HACS [53]. We select videos and objects based on the following criteria:

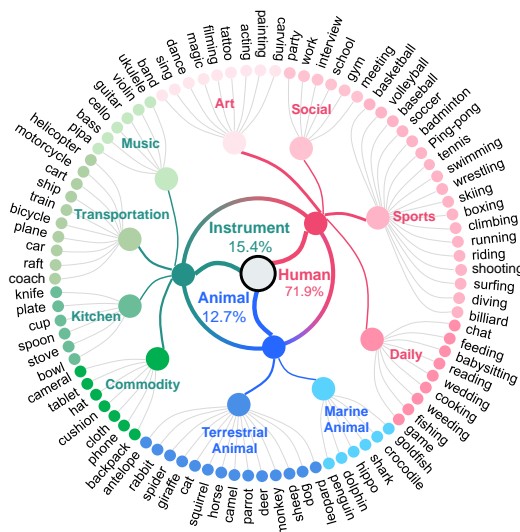

Figure 3: Overview of object categories and scenes in Long-RVOS.

- The video duration exceeds 20 seconds.
- Objects that belong to background, ambiguous or unknown categories are excluded.
- Each selected video must contain more than two valid objects, and at least one object is not continuously visible.

With these criteria, we have initially collected over 3K videos and 8K objects as candidates. After careful inspections on quality, we finally select 2,193 videos and 6,703 objects to build Long-RVOS.

### 3.2 Dataset Annotation.

**Text Annotation.** We develop an online platform for annotating object descriptions. This platform randomly samples a video from our dataset and displays it, with all target objects highlighted by bounding boxes. To ensure the diversity of annotations, each video can be sampled repeatedly at most three times. The annotators consisting of 20 college students are asked to watch the videos and provide the following three types of descriptions for each object:

- **Static type** includes appearance (e.g., colors and shapes), relative position (e.g., "the left cat"), and environmental context (e.g., "on the grass").
- **Dynamic type** includes motions, changes over time (e.g., in position or state) and interactions with other entities (e.g., "the cat chasing a mouse").
- **Hybrid type** integrates both static and dynamic attributes to provide comprehensive object cues.

The key annotation principle is that every single description, regardless of type, must clearly distinguish the target object from others. For objects that cannot be distinguished by only static or dynamic attributes, the corresponding type of annotation can be skipped. After this annotation phase, we have collected over 30K text descriptions. These annotations and the corresponding videos are then sent to a validation team formed by three experts for quality verification. Any descriptions that violate our principle are directly removed. Besides, we do not use techniques like synonym replacement to artificially scale up the text annotations, keeping the dataset clear and authentic to support reliable RVOS training. Finally, we gather 24,689 high-quality descriptions for building Long-RVOS.

**Mask Annotation.** Our source datasets [6, 12, 37] have provided sparse bounding-box annotations. For each object, we segment the video into clips based on the annotated frames. Then, we utilize SAM2 [35], the state-of-the-art VOS model, to track the objects within each clip and produce high-quality masks, by regarding the annotated bounding box as the first-frame prompt. To ensure

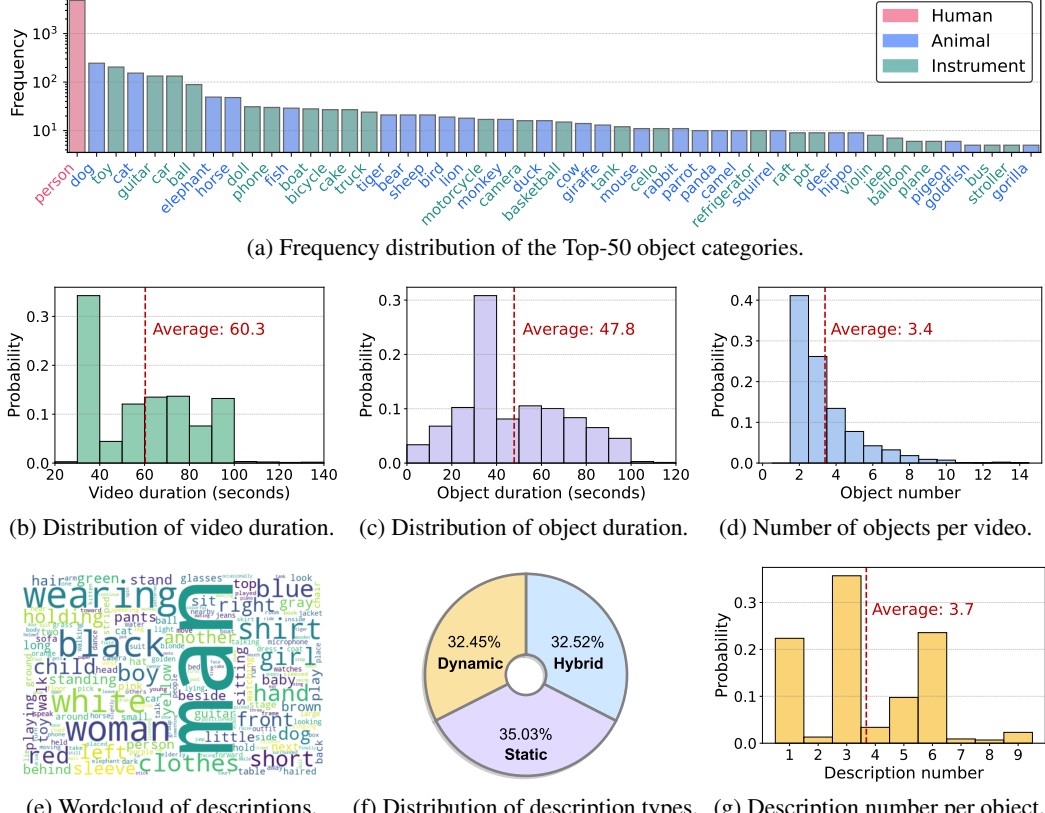

(a) Frequency distribution of the Top-50 object categories.

(b) Distribution of video duration.

(c) Distribution of object duration.

(d) Number of objects per video.

(e) Wordcloud of descriptions.

(f) Distribution of description types.

(g) Description number per object.

Figure 4: Representative statistics of Long-RVOS.

annotation quality, we conduct an iterative *check–correct* workflow. Specifically, the validation team checks every object's mask separately in the video, and marks the objects with inaccurate annotations. To facilitate the correction process, we develop an interactive annotation tool based on SAM2. This tool loads a marked object each time and visualizes its masks in the video. Nine annotators use our tool to refine the masks with point or box prompts, and remove masks from object-absent frames. The corrected results are then returned to the checking queue, and this *check–correct* loop repeats until all mask annotations are qualified.

## 3.3 Dataset Statistics

A detailed comparison with five existing RVOS datasets is shown in Table 1. Notably, Long-RVOS offers significantly longer video duration than existing datasets. In addition, it contains the largest number of object classes and mask annotations. The large scale of Long-RVOS supports comprehensive training and evaluation of RVOS models.

**Diverse Objects and Scenes.** Long-RVOS is constructed by integrating multiple sources of video datasets, achieving a wide variety of objects and scenes, as illustrated in Figure 3. These sources include indoor videos from Charades [38], outdoor videos from LaSOT [9], movie scenes from AVA [13], egocentric videos from Ego-Exo4D [12], and more diverse videos from other datasets [37, 41, 53]. In total, Long-RVOS contains 163 object categories, significantly surpassing the existing RVOS datasets. As shown in Figure 4 (a), while Long-RVOS primarily focuses on human instances (71.9%), it also covers a diverse range of animals (12.7%) and instruments (15.4%). In Figure 4 (b)-(d), we present further statistics on the videos and objects in Long-RVOS. Notably, the object number of each video spans from 2 to 14, preventing over-reliance on the most salient object and highlighting text-guided segmentation. With such extensive diversity, Long-RVOS can serve a comprehensive benchmark for RVOS research, facilitating the development of more real-world applications.

**Diverse Descriptions.** In real-world applications, user queries are always unpredictable. They might refer to salient attributes or instantaneous actions. To enable more comprehensive evaluation of model capabilities, Long-RVOS introduces three distinct types of text descriptions — *static*, *dynamic*, and *hybrid*. By explicitly categorizing these types, Long-RVOS prevents evaluation bias toward specific attribute cues (e.g., color or position), ensuring a fair and robust assessment. We present the detailed statistics of text descriptions in Figure 4 (c)-(g). Critically, Long-RVOS maintains a balanced distribution of text types, and the description number for each object can vary from 1 to 9. These properties encourage comprehensive learning of diverse object attributes. With its explicit type annotations and diverse object descriptions, Long-RVOS provides a comprehensive benchmark for training and evaluating RVOS models in more realistic scenarios.

## 3.4 Evaluation Metrics

Previous RVOS benchmarks tend to evaluate model performance with the frame-wise spatial metrics, such as $\mathcal{J}\&\mathcal{F}$. Here, $\mathcal{J}$ denotes the Intersection-over-Union (IoU) between the predicted and ground-truth masks, $\mathcal{F}$ measures the contour accuracy, and $\mathcal{J}\&\mathcal{F}$ is their average over all the frames. However, these metrics focus solely on the per-frame segmentation quality, neglecting the temporal consistency. A robust RVOS model should accurately segment the target when it is present and correctly output an empty mask when it is absent. Inspired by the field of spatiotemporal video grounding [40, 52], we additionally introduce two new metrics, tIoU and vIoU, in Long-RVOS to individually evaluate the temporal and spatiotemporal performance.

Formally, let $\hat{M}_t, M_t \in \{0,1\}^{H \times W}$ denote the predicted and ground-truth masks at $t$-th frame, respectively, where $t \in [1, T]$. The frame-index sets of non-empty masks are defined as $\hat{\mathcal{T}} = \{t \mid \|\hat{M}_t\|_0 > 0\}$ (for predictions) and $\mathcal{T} = \{t \mid \|M_t\|_0 > 0\}$ (for the ground-truth), where the $\ell_0$-norm $\|\cdot\|_0$ denotes the count of non-zero elements. Then, tIoU is obtained by computing their IoU:

$$\text{tIoU} = \frac{T_i}{T_u}, \quad \text{where } T_i = \hat{\mathcal{T}} \cap \mathcal{T} \text{ and } T_u = \hat{\mathcal{T}} \cup \mathcal{T}, \tag{1}$$

and vIoU computes the volume IoU between predicted and ground-truth mask sequences:

$$\text{vIoU} = \frac{1}{T_u} \sum_{t \in T_i} \mathcal{J}_t, \quad \text{where } \mathcal{J}_t = \frac{\hat{\mathcal{M}}_t \cap \mathcal{M}_t}{\hat{\mathcal{M}}_t \cup \mathcal{M}_t}. \tag{2}$$

By combining the spatial metric $\mathcal{J}\&\mathcal{F}$, temporal metric tIoU and spatiotemporal metric vIoU, Long-RVOS establishes a rigorous evaluation protocol for RVOS research.

## 4 ReferMo: A Baseline Approach

As illustrated in Figure 5, ReferMo decomposes the video into a sequence of clips, each consisting of a high-resolution keyframe and subsequent low-resolution motion frames. Then, it perceives the static appearance and short-term motion within local video clip, and captures the object target in long-term context by integrating the cross-clip information. Critically, ReferMo only predicts target masks over the keyframes, and the masks on the remain frames are generated by a pretrained object tracker (e.g., SAM2 [35]). In this way, ReferMo achieves a trade-off between training costs and long-term understanding without processing a large number of high-resolution frames.

### 4.1 Video Decomposition

Typically, a long-term video is composed of multiple shots, and the video frames within each shot often show significant temporal redundancy. This redundancy can be efficiently described by motion information to reduce the frame-by-frame computations. Inspired by Video-LaVIT [18], we employ the MPEG-4 [23] compression technique to extract keyframe and motion information from the videos. More sophisticated (but expensive) keyframe selection strategies [42, 45] can also be explored, but they are not the primary focus of this work. In MPEG-4, a video is decomposed into multiple clips, where each clip consists of a keyframe $\mathcal{I} \in \mathbb{R}^{H \times W \times 3}$ and the motion vectors $\mathcal{M} \in \mathbb{R}^{T \times \frac{H}{16} \times \frac{W}{16} \times 2}$ of its subsequent $T$ frames. Unlike the dense optical flow, these motion vectors can be directly extracted during the compressed video decoding process, making them well-suited for processing large-scale, long-term videos. The details of motion extraction process are provided in the supplementary.

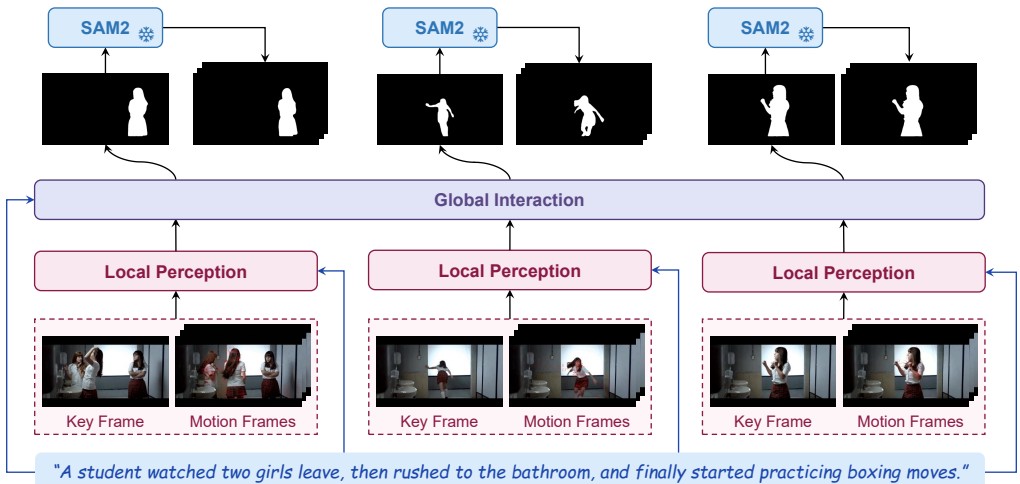

Figure 5: Overview of ReferMo. A video is decomposed into clips (keyframe + motion frames). ReferMo perceives the static attributes and short-term motions within each clip, then aggregates inter-clip information capture the global target. Notably, ReferMo is supervised by only keyframe masks, and SAM2 is only used at inference for target tracking in subsequent frames.

## 4.2 From Local Perception to Global Interaction

Different from the previous RVOS methods [25, 30, 48] that perform vision-language fusion on each single frame, we introduce motion representations to enable clip-level vision-language fusion. For each video clip, as shown in Figure 6, the local perceiver encodes the text, keyframe and motion information through three separate encoders, and then employs a multi-modal fuser to progressively aggregate these information for clip-level object extraction. By collecting the objects across different video clips, we perform global temporal interaction to enable consistent object prediction and long-term temporal understanding.

**Motion Encoder.** The motion vectors are first embedded into a $d$-dimensional space via a linear projector. Then, the motion encoder performs self-attention separately along the spatial and temporal dimensions to extract the spatiotemporal motion features $M \in \mathbb{R}^{T \times \frac{H}{16} \times \frac{W}{16} \times d}$. Notably, we implement the spatial attention as deformable attention due to the large number of spatial tokens.

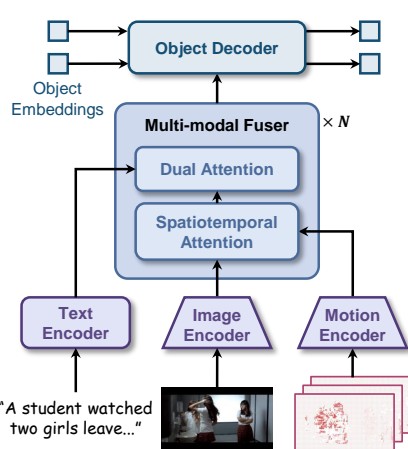

Figure 6: Overview of local perceiver.

**Image-Motion Fusion.** Modern image encoders (e.g., Swin Transformer [29]) typically output multi-scale feature maps $I_i \in \mathbb{R}^{H_i \times W_i \times d}$, $i \in [1, 4]$. To match these spatial resolutions, we adopt a series of spatial convolutions with specific strides over the motion features $M$ to produce multi-scale motion features $M_i \in \mathbb{R}^{T \times H_i \times W_i \times d}$. At each scale $i$, we treat the keyframe feature $I_i$ as *query* and perform cross-attention along the temporal dimension to aggregate $M_i$ into $\widetilde{M_i} \in \mathbb{R}^{H_i \times W_i \times d}$. To avoid undesired motion noise, we fuse the keyframe and motion features via the spatial-aware and channel-aware gating mechanisms:

$$M_i^* = (\underbrace{\sigma(I_i \cdot W_{down}^I)}_{\text{Spatial Gate}} \odot (\widetilde{M_i} \cdot W_{down}^M)) \cdot W_{up}, \tag{3}$$

$$F_i = I_i + \underbrace{\gamma_i}_{\text{Channel Gate}} \odot \max(M_i^*, 0)^2, \tag{4}$$

where $W_{down}^I, W_{down}^M \in \mathbb{R}^{d \times r}$ indicate the low-rank projectors that compress the features to a lower dimension $r$, and $W_{up} \in \mathbb{R}^{r \times d}$ is a projector to resort the dimension. $\sigma$ denotes Sigmoid function and $\odot$ denotes Hadamard product. $\gamma \in \mathbb{R}^d$ is a learnable vector to modulate the channel-wise weights.

**Vision-Language Fusion.** We use the dual cross-attention modules [24, 28] for deep vision-language fusion. Formally, given the clip-level vision features $F \in \mathbb{R}^{N \times d}$ and the language features $E \in \mathbb{R}^{L \times d}$, where $N$ and $L$ individually denote their token number, we derive the cross-modal enhanced vision features $\widetilde{F}$ and language features $\widetilde{E}$ as follows:

$$\mathcal{A} = \frac{F \cdot E^\top}{\sqrt{d}}, \qquad \widetilde{F} = \text{Softmax}(\mathcal{A}) \cdot E, \qquad \widetilde{E} = \text{Softmax}(\mathcal{A}^\top) \cdot F. \tag{5}$$

For simplicity, the linear projections for multi-head attentions are omitted. The output features $\widetilde{F}$ and $\widetilde{E}$ are then fed into the object decoder to extract object features.

**Global Interaction.** To enable consistent object prediction and long-term temporal understanding, we collect the object features across video clips to perform global temporal interactions. Following ReferDINO [25], we use the Hungarian algorithm [22] to align the objects clip-by-clip. Then, we perform temporal self-attention over the aligned object features to achieve global modeling. For better modality alignment, we also infuse the language information $\widetilde{E}$ into the object features through a cross-attention layer. Finally, the interacted object features are output to the segmentation head for generating instance masks. Note that these masks are only predicted for the key frame within each clip, serving as object anchors for SAM2's mask propagation in subsequent frames.

## 5 Experiments

### 5.1 Experiment Setup

**Dataset Split.** Long-RVOS is a large-scale dataset containing 2,193 videos and 24,689 sentences, which are split into three subsets: a training set of 1,855 videos and 20,722 sentences, a validation set of 113 videos and 1,379 sentences, and a test set of 225 videos and 2,588 sentences.

**Evaluation Metrics.** We use three kinds of evaluation metrics: the spatial metric $\mathcal{J}\&\mathcal{F}$, the temporal metric tIoU and the spatiotemporal metric vIoU. Long-RVOS provides three types of descriptions: *static*, *temporal* and *hybrid*. We report performance for each type separately and overall. Additionally, we report the FPS for each competitor because efficiency is a major concern for long-video processing.

**Implementation Details.** We follow the default hyper-parameter settings of ReferDINO [25] and use Swin-Tiny as the backbone. For SAM2 [35], we use the sam2.1_hiera_large version. In MPEG-4 [23], each video clip typically consists of a keyframe and the motion vectors for up to 11 subsequent frames. During training, we randomly sample 6 clips and use 3-frame motion vectors. The input frames are resized to have the longest side of 640 pixels and the shortest side of 360 pixels during training and evaluation. Following the settings on MeViS [7], we do not use referring image segmentation datasets (e.g., RefCOCO/+/g [19, 32]) for pretraining. We train ReferMo on Long-RVOS dataset for 6 epochs, which take 24 hours on 8 Nvidia A6000 GPUs.

### 5.2 Benchmark Results

**Overall Comparison.** We compare ReferMo with six recent RVOS methods on Long-RVOS. All models in comparison are trained on Long-RVOS under consistent experimental settings for fairness. As demonstrated in Table 2, realistic long-video scenarios remain a significant challenge for current RVOS models. While the SAM2-based methods [5, 26] achieve SOTA performance on existing short-term benchmarks [7, 20, 36], they significantly struggle in Long-RVOS. This suggests that their improvements may primarily stem from SAM2's superior tracking and segmentation capabilities, rather than better language-guided object understanding. As the videos grow longer and more complex, it becomes more challenging to perform video-language reasoning and distinguish the objects, which leads to their performance degradation. In contrast, our baseline ReferMo integrates the static attributes, short-term dynamics and long-term dependencies to perform object-level visual-language reasoning, achieving significant improvements over existing methods. These findings highlight the need for both frame-level segmentation precision and video-level visual-language understanding to address the long-video challenges in Long-RVOS.

**Fine-grained Evaluation.** Long-RVOS provides three types of text descriptions to enable rigorous evaluation. For most models, the performance for static and hybrid types is comparable and largely better than that for dynamic type. This implies a strong bias in current RVOS models toward static

| Method | Year | Static | | | Dynamic | | | Hybrid | | | Overall | | | FPS |
|---|---|---|---|---|---|---|---|---|---|---|---|---|---|---|
| | | $\mathcal{J}\&\mathcal{F}$ | tIoU | vIoU | $\mathcal{J}\&\mathcal{F}$ | tIoU | vIoU | $\mathcal{J}\&\mathcal{F}$ | tIoU | vIoU | $\mathcal{J}\&\mathcal{F}$ | tIoU | vIoU | |
| *Without SAM / SAM2* | | | | | | | | | | | | | | |
| SOC [30] | 2023 | 34.8 | 67.7 | 28.4 | 34.9 | 68.7 | 28.8 | 35.1 | 68.0 | 28.5 | 34.9 | 68.1 | 28.6 | 53.8 |
| MUTR [48] | 2024 | 43.0 | 70.1 | 36.7 | 40.2 | 70.8 | 34.8 | 43.2 | 70.3 | 37.2 | 42.2 | 70.4 | 36.2 | 20.4 |
| ReferDINO [25] | 2025 | 50.7 | **71.9** | 42.8 | 45.9 | **71.9** | 38.9 | 49.2 | **71.5** | 41.7 | 48.7 | **71.7** | 41.2 | 46.4 |
| *With SAM / SAM2* | | | | | | | | | | | | | | |
| VideoLISA [1] | 2024 | 34.3 | 69.6 | 28.9 | 31.0 | 69.7 | 26.9 | 33.9 | 69.4 | 28.6 | 33.1 | 69.6 | 28.2 | 6.6 |
| GLUS [26] | 2025 | 36.4 | 68.2 | 34.3 | 37.6 | 68.9 | 35.8 | 35.9 | 68.0 | 33.9 | 36.6 | 68.4 | 34.6 | 3.6 |
| SAMWISE [5] | 2025 | 36.6 | 68.4 | 29.2 | 34.3 | 68.6 | 28.1 | 33.8 | 69.4 | 28.4 | 35.6 | 68.4 | 28.6 | 7.0 |
| **ReferMo** | 2025 | **53.5** | 71.4 | **44.0** | **48.1** | 71.2 | **40.1** | **52.2** | 71.2 | **43.6** | **51.3** | 71.2 | **42.6** | 52.5 |

Table 2: Comparison of state-of-the-art RVOS models on Long-RVOS test set. FPS is estimated at 360P on Nvidia A6000 GPUs, excluding the video loading time.

| Dataset | | Point | Box | Mask |
|---|---|---|---|---|
| MeViS [7] | Valid_u | 77.3 | 80.0 | 80.6 |
| Long-RVOS | Valid | 53.4 | 54.5 | 53.5 |
| | Test | 52.8 | 53.9 | 53.3 |

(a) Oracle analysis with SAM2.

| Model | $\mathcal{J}\&\mathcal{F}$ | $\mathcal{J}$ | $\mathcal{F}$ |
|---|---|---|---|
| ReferDINO | 49.1 | 47.6 | 50.6 |
| ReferMo | 49.6 | 48.0 | 51.2 |
| -w/o motion | 47.5 | 46.0 | 48.9 |

(b) Results on the keyframes.

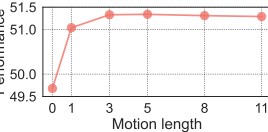

(c) Different motion lengths.

Table 3: Oracle analysis and ablation studies.

attributes. Across different models, while the $\mathcal{J}\&\mathcal{F}$ scores show significant variance, their tIoU are relatively consistent. This reveals that existing RVOS models have little performance gap in temporal consistency, highlighting the need for effective tracking mechanisms to handle frequent target disappearance in long-term videos. ReferMo significantly outperforms other models across various types and metrics, except for tIoU, where it is slightly inferior to ReferDINO. We speculate that this is because ReferMo only performs language-guided reasoning on keyframes, resulting in suboptimal object identification on motion frames.

**Oracle Analysis.** We provide SAM2 with first-frame ground-truth object prompts and evaluate its tracking performance across different datasets. As shown in Table 3 (a), the oracle results for Long-RVOS (52.8~54.5 $\mathcal{J}\&\mathcal{F}$) are significantly lower than those for MeViS (77.3~80.6 $\mathcal{J}\&\mathcal{F}$). The notable performance gap of nearly 25% demonstrates the long-term challenges in Long-RVOS.

## 5.3 Ablation Studies

**Results on Keyframes.** In Table 3 (b), we compare the performance of ReferMo and ReferDINO [25] on the keyframes. We focus on the spatial metrics since the length of the keyframe sequence is short. Note that ReferDINO is trained on all frames, while our ReferMo is only trained on keyframes. However, ReferMo still outperforms ReferDINO by 0.5% in $\mathcal{J}\&\mathcal{F}$, owing to the integration of motion information. When ablating it, we see a significant 2.1% performance drop in $\mathcal{J}\&\mathcal{F}$. These results encourage further exploration of sparse-frame supervision for RVOS task.

**Effect of Motion Information.** We investigate the impact of varying the number of motion frames in ReferMo. As shown in Table 3 (c), the performance without motions is only 49.7 $\mathcal{J}\&\mathcal{F}$. However, even using just one motion frame yields +1.6% $\mathcal{J}\&\mathcal{F}$ improvements. Increasing the motion length to 3 frames improves $\mathcal{J}\&\mathcal{F}$ to 51.3, but further increasing only leads to marginal gains.

## 6 Conclusion

In this work, we introduce Long-RVOS, a large-scale benchmark for long-term referring video object segmentation, comprising over 2,000 videos averaging 60+ seconds to address the limitations of existing short-term datasets. To enable comprehensive and rigorous evaluation, we provide three types of descriptions and two novel metrics, tIoU and vIoU. Results on Long-RVOS indicate that current RVOS methods struggle severely in long-video scenarios. Furthermore, we propose ReferMo, a simple motion-enhanced baseline that significantly outperforms existing SOTA methods on long-term videos. We believe that Long-RVOS and ReferMo will provide a foundation for future research to develop robust RVOS models tackling real-world long-form videos.

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
