# OpenReview forum: "Long-RVOS: A Comprehensive Benchmark for Long-term Referring Video Object Segmentation"
_NeurIPS.cc/2025/Conference — Submitted to NeurIPS 2025_

### Official Review · Reviewer_okyn · 2025-06-15

**Clarity:** 3
**Significance:** 2
**Originality:** 3
**Rating:** 4
**Confidence:** 5

**Summary:**

This paper collects a new dataset introducing long videos (average >60s) for RVOS. A straightforward baseline is proposed along with new metics evaluating temporal and spatiotemporal consistency.

**Questions:**

see above

**Ethical Concerns:**

["NO or VERY MINOR ethics concerns only"]

**Final Justification:**

My concerns are addressed. I have no further comment. I decide to raise my score.

**Limitations:**

yes

**Quality:**

3

**Strengths And Weaknesses:**

Strengths
* The introduction of Long-RVOS addresses a critical gap by focusing on long videos (average >60s), moving beyond the short-clip paradigm prevalent in existing RVOS datasets.
* Taxonomy of videos (unfortunately not for texts) are detailed and structured.

Concerns
* While the paper effectively highlights the visual challenges of long videos (occlusion, reappearance, shot changes), it provides less in-depth analysis of the specific linguistic and cross-modal alignment difficulties introduced by long-term contexts. How do long descriptions, temporal references ("later", "after the cut"), or complex event sequences impact performance? This aspect deserves more exploration.
* The observation that current SOTA methods "struggle severely" is noted, but the paper could delve deeper into why they fail beyond the implied lack of long-term modeling. Is it primarily due to limited temporal context windows, failure to track through occlusions/reappearances, difficulty resolving ambiguous language over time, or a combination?
* This paper's biggest contribution is the new dataset. While authors choose not to submit this paper to dataset track, which has extra rules, like requiring authors to submit the dataset and the code used to test it. This ensures everyone can check the data is good and rerun the tests. Since this paper wasn't submitted that way, I worry we can't fully trust the dataset quality or easily reuse it.
* The solution they propose works well for long videos. That's good. But the ideas inside it – using motion info and combining short-term/local info with longer-term/global info – are solutions we've seen used in other video problems before. It's a good baseline, but it doesn't introduce any major new tricks or concepts.
* Using metrics to check consistency over time isn't actually new outside of RVOS; it's common in other video tasks. While this is the first time to bring temporal and spatiotemporal concistency metrics into RVOS, it feels more like catching up to what other fields already do, rather than inventing something truly novel.

---

> ### Author Rebuttal · Authors · 2025-07-30
>
> Thank you for the constructive comments. We provide our responses as follows.
>
> > **Q1. How do long descriptions, temporal references, or complex event sequences impact performance?**
>
> * **Regarding long descriptions.** We evaluate the impact of varying description lengths and present the results in Table R1. As description length increases, slight performance declines are observed across models. However, our ReferMo consistently outperforms existing methods across different description lengths.
>
> |Methods|<10|[10,20]|>20|
> |:-|:-:|:-:|:-:|
> |MUTR|42.4|41.4|40.3|
> |SAMWISE|43.9|43.1|41.9|
> |ReferDINO|49.9|49.0|46.3|
> |**ReferMo (Ours)**|**51.7**|**51.6**|**50.4**|
> |
>
> Table R1. Overall J&F results for different description lengths.
>
> * **Regarding temporal references.** In Lines 308-311, we demonstrated that most RVOS models exhibit a strong bias toward static attributes, with significantly weaker performance on Dynamic types (involving motion and temporal elements).
>
> * **Regarding complex event sequences.** We categorized the samples into *single-event*, *two-event*, and *multi-event* groups based on the keywords (e.g., *then*, *finally*, *ultimately*) in descriptions. As shown in Table R2, performance across models declines as the event number increases, yet our ReferMo consistently outperforms existing methods. Also, these results highlight the significance of our long-term benchmark for evaluating the capabilities of RVOS models in understanding complex event sequences.
>
>
> |Methods|*single-event*|*two-event*|*multi-event*|
> |:-|:-:|:-:|:-:|
> |MUTR|41.1|40.6|39.8|
> |SAMWISE|42.8|40.5|41.6|
> |ReferDINO|48.2|45.6|42.6|
> |**ReferMo (Ours)**|**50.8**|**48.3**|**45.5**|
> |
>
> Table R2. Overall J&F results by event complexity.
>
> Thanks for your suggestions and we will include them in our final version.
>
> > **Q2. Why SOTA methods fail in Long-RVOS.**
>
> The reasons are multifaceted. We'd like to clarify that, in Introduction (Lines 37-46), we discussed the limitations of current SOTA methods in handling long-video scenarios, including their inefficiency in integrating long-term information and capturing fleeting cues, and the training-inference gap due to GPU limitation. Additionally, in Section 5.2 (Lines 295-317), we empirically demonstrated the insufficient vision-language understanding in recent SAM2-based models, the potential static bias, and ineffective long-term consistency. These discussions and analyses can provide clear insights and are far from simply "*implying the lack of long-term modeling*".
>
> To further address your concern, we present results at various object occlusion rates in Table R3. As the occlusion rate increases, RVOS models exhibit significant performance declines, though the decline extent varies across models. In comparison, our ReferMo model remains robust in most cases (with occlusion rates ranging from 0 to 0.75) but struggles in extremely high-occlusion scenarios (>0.75), as it relies solely on the keyframes. These findings also highlight the significance of our benchmark, which includes much more occlusion scenarios than existing benchmarks, as demonstrated in Table 5 of the Appendix.
>
> |Methods|[0,0.25]|[0.25,0.5)|[0.5,0.75)|[0.75,1]|
> |:-|:-:|:-:|:-:|:-:|
> |SOC|39.8|32.8|28.6|15.2|
> |MUTR|50.1|39.4|30.5|9.5|
> |ReferDINO|54.7|46.0|41.8|13.7|
> |VideoLISA|40.2|30.1|22.1|10.5|
> |GLUS|49.1|31.0|18.5|4.1|
> |SAMWISE|46.7|42.0|39.8|**24.2**|
> |**ReferMo (Ours)**|**57.3**|**48.7**|**44.6**|20.6|
> |
>
> Table R3. Overall J&F results at various object occlusion rates.
>
> > **Q3. Why not submit to dataset track.**
>
> We'd like to clarify that we chose the Main Conference track because, in addition to introducing a new dataset, our work establishes a promising and effective baseline method for long-term RVOS. We also confirmed that our submission completely complies with the NeurIPS rules. Furthermore, we will definitely release the dataset and code to enable community verification and use, as committed in the paper.
>
> > **Q4. It's a good baseline, but it doesn't introduce new tricks or concepts.**
>
> We appreciate your recognition of our method's effectiveness for the new challenge. However, we'd argue that the novelty of our method lies not in inventing individual components (e.g., motion information and local-global structure), but in establishing **a new paradigm for tackling long-term RVOS**.
>
> 1. Our method decouples RVOS into keyframe object identification and non-keyframe mask propagation, enabling efficient training and inference on realistic long-form videos. This is meaningful and significantly different from existing methods that perform dense cross-modal reasoning on every video frame.
>
> 2) We propose introducing low-res motion frames to bridge the training-inference gap in temporal receptive fields. This solution has not been studied before and is far from straightforward. It can also provide insights for other video fields that require dense computations (e.g., video super-resolution and long-video generation).
>
> Combining these elements, we believe the novelty of our method and its value to the field have been demonstrated.
>
> > **Q5. The metrics are not newly invented.**
>
> We respectfully argue that while the two metrics are established in other video tasks, their first application to RVOS field is still novel and significant. As noted in Lines 47-53, current benchmarks focus mainly on spatial metrics and overlook the temporal/spatiotemporal evaluation. By introducing these metrics to RVOS, we address a long-standing gap in the field. Besides, our work consistently aims to advance the RVOS field towards more practical scenarios, rather than "inventing" some completely new metrics.

---

> > ### Comment · Reviewer_okyn · 2025-08-06
> >
> > Thank you for the thorough response. I am convinced that the proposed baseline is strong enough for publication. I decide to raise my score. I may still suggest the authors considering dataset track if possible.

---

> > > ### Author Response · Authors · 2025-08-06
> > >
> > > We sincerely appreciate your positive feedback and your recognition of our method. We are truly encouraged by your decision to raise your score. We also thank you for the suggestion regarding the dataset track, which we will carefully consider for future work.

---

> ### Comment · Area_Chair_BqhG · 2025-08-05
> **Update after rebuttal**
>
> Dear Reviewer,
>
> The authors’ rebuttal has been posted. Please check the author's feedback, evaluate how it addresses the concerns you raised, and post any follow-up questions to engage the authors in a discussion. Please do this ASAP.
>
> Thanks.
>
> AC

---

### Official Review · Reviewer_4DtK · 2025-06-25

**Clarity:** 4
**Significance:** 3
**Originality:** 3
**Rating:** 5
**Confidence:** 5

**Summary:**

This paper introduces Long-RVOS, the first large-scale benchmark specifically designed for long-term referring video object segmentation (RVOS). The dataset features over 2,000 minute-long videos annotated with diverse object categories and three types of language descriptions (static, dynamic, hybrid). To evaluate performance more rigorously, the authors propose two new metrics, tIoU and vIoU, capturing temporal and spatiotemporal consistency beyond frame-wise accuracy. The paper also presents a baseline model, ReferMo, which integrates motion information into a local-to-global architecture, using sparse keyframe supervision combined with SAM2-based propagation at inference time.
Experiments show that ReferMo achieves strong performance, outperforming existing methods on Long-RVOS.

**Questions:**

1. How does ReferMo handle cases where the target object is absent in the keyframe?

2. To what extent does SAM2 contribute to the final performance during inference?
Given that SAM2 is used to propagate masks beyond the predicted keyframes, could the authors provide an ablation or alternative comparison to isolate the contribution of ReferMo from that of SAM2? For example, what happens if other propagation strategies are used?

3. Were the reported results for SAMWISE obtained using the latest version of the codebase?
Recent updates to the SAMWISE repository indicate that a bug affecting inference quality was present in earlier releases. Could the authors clarify whether the SAMWISE results reported in the paper reflect the corrected implementation? If the reported results are based on an outdated version, it would be helpful to update the comparison using the corrected implementation for a fair evaluation.

**Ethical Concerns:**

["NO or VERY MINOR ethics concerns only"]

**Final Justification:**

I remain convinced of my initial positive assessment. The proposed Long-RVOS dataset fills an important gap in RVOS research: existing benchmarks are typically limited to short clips and are insufficient for evaluating long-term reasoning. I believe Long-RVOS will be a valuable and widely used resource for the community. The proposed baseline, ReferMo, is simple yet competitive, and the additional experiments in the rebuttal clarify my remaining doubts.

**Limitations:**

yes

**Paper Formatting Concerns:**

No major formatting issues.

**Quality:**

3

**Strengths And Weaknesses:**

**Strenghts**

**First large-scale benchmark for Long-Term RVOS**
The paper introduces Long-RVOS, the first benchmark specifically designed for evaluating referring video object segmentation in minute-long, realistic video scenarios. Compared to existing datasets limited to short clips, Long-RVOS features significantly longer durations, diverse object types, and multiple instances per video, making it a valuable resource for practical settings.


**Explicit annotation of text types for fine-grained evaluation**
 Each object is annotated with three distinct types of natural language descriptions (static, dynamic, hybrid), allowing for fine-grained evaluation of a model ability to understand different semantic cues. This is a novel addition that enables more precise performance analysis and avoids overfitting to static or appearance-based prompts.


**Strong baseline (ReferMo) and extensicve evaluation**
 The proposed ReferMo introduces a local-to-global architecture that integrates appearance, motion, and temporal reasoning across clips. The paper benchmarks six state-of-the-art methods and performs detailed ablations. Oracle experiments using SAM2 highlight the difficulty of the benchmark and confirm the need for improved long-term reasoning. Despite its simplicity, ReferMo outperforms several strong baselines, including SAM-based models, in long-video settings.


**Weaknesses**

1. The model relies on a single keyframe per clip for language-guided mask prediction, with subsequent frames handled via mask propagation. In long and complex videos where the target object is frequently occluded, it is possible that multiple keyframes fail to capture the object, if occluded. While efficient, this design introduces a clear failure mode when visibility is limited across clips.

2. While the proposed method achieves strong performance, it inherently relies on access to future frames (e.g., for inter-clip aggregation). This design limits its applicability in streaming scenarios, where frames must be processed as they arrive. Although not a flaw per se, this aspect could benefit from further discussion, especially in contrast with recent RVOS models that operate in a streaming fashion.

---

> ### Author Rebuttal · Authors · 2025-07-30
>
> Thank you for your appreciation of our work and insightful suggestions. We provide our responses as follows.
>
> > **Q1. The version of SAMWISE.**
>
> Thank you for pointing it out. SAMWISE is an impressive work in adapting SAM2 for RVOS. We used the commit 08eb4ed (Apr 7), which was the latest version before the NeurIPS deadline (May 15). Following your suggestions, we re-evaluate SAMWISE with the latest codebase (Jun 28) and achieve higher results, as shown in Table R1. Due to formatting limitations in markdown tables, more detailed results will be updated in the final version.
>
> Please note that all results for SAMWISE in this rebuttal are evaluated with the latest version.
>
> |Dataset|Static|Dynamic|Hybrid|Overall|
> |:-:|:-:|:-:|:-:|:-:|
> |Valid|45.7|39.5|44.4|43.3|
> |Test|44.4|41.1|43.7|43.1|
> |
>
> Table R1. J&F Results of SAMWISE with latest codebase.
>
> > **Q2. How does ReferMo handle cases where the target object is absent in the keyframe?**
>
> We agree that this is a common challenge for keyframe-based methods. When the target object is absent in a keyframe, ReferMo is expected to predict an empty mask for SAM2, resulting in an empty mask sequence for that clip. To address your concern, we present the results under varying object occlusion rates in Table R2. Despite relying soly on keyframe reasoning, ReferMo demonstrates significant performance advantages in most occlusion cases (0 to 0.75). Even in extremely high-occlusion cases (>0.75), it still outperforms most existing methods, except for SAMWISE.
>
> Therefore, although it depends solely on keyframes, ReferMo remains sufficiently robust in most non-extreme cases. We sincerely thank you for raising this point. We will include these results in our final version and make further efforts to address this challenge in future work.
>
> |Methods|[0,0.25]|[0.25,0.5)|[0.5,0.75)|[0.75,1]|
> |:-|:-:|:-:|:-:|:-:|
> |SOC|39.8|32.8|28.6|15.2|
> |MUTR|50.1|39.4|30.5|9.5|
> |ReferDINO|54.7|46.0|41.8|13.7|
> |VideoLISA|40.2|30.1|22.1|10.5|
> |GLUS|49.1|31.0|18.5|4.1|
> |SAMWISE|46.7|42.0|39.8|**24.2**|
> |**ReferMo (Ours)**|**57.3**|**48.7**|**44.6**|20.6|
> |
>
> Table R2. Overall J&F results at various object occlusion rates.
>
> > **Q3. Potential applicability in streaming scenarios.**
>
> Thanks for your suggestions. Extending RVOS to streaming scenarios is indeed valuable. Intuitively, ReferMo can be adapted to streaming videos by replacing global temporal attention with causal attention. However, an additional post-correction mechanism is also crucial for object correction. We will add further discussions (including comparisons with streaming models like SAMWISE) in the final version.
>
>
> > **Q4. Ablation with other propagation strategies.**
>
> We follow your suggestion to replace SAM2 with other propagation models (e.g., Xmem++[R1] and Cuite[R2]) in Table R3, which shows that SAM2 contributes 1.1-2.5% J&F gains to overall performance. Notably, even when combined with these traditional propagation models, our ReferMo still outperforms existing SAM2-based RVOS methods, validating the robustness of our approach.
>
> |Method|Static|Dynamic|Hybrid|Overall|
> |:-|:-:|:-:|:-:|:-:|
> |**SAM2-based Methods**
> |GLUS|36.4|37.6|35.9|36.6|
> |SAMWISE|44.4|41.1|43.7|43.1|
> |**ReferMo (Ours)**
> |+ Xmem++[R1]|50.8|45.8|49.7|48.8|
> |+ Cuite[R2]|52.2|47.0|51.1|50.2|
> |+ SAM2|53.5|48.1|52.2|51.3|
> |
>
> Table R3. J&F comparison of ReferMo variants and baselines.
>
> [R1] Bekuzarov, Maksym, et al. "Xmem++: Production-level video segmentation from few annotated frames." ICCV, 2023.
>
> [R2] Cheng, Ho Kei, et al. "Putting the object back into video object segmentation." CVPR, 2024.

---

> > ### Comment · Reviewer_4DtK · 2025-08-04
> >
> > Thank you to the authors for the detailed rebuttal. I have also carefully read the other reviews and the replies.
> >
> > I remain convinced of my initial positive assessment. The proposed Long-RVOS dataset fills an important gap in RVOS research: existing benchmarks are typically limited to short clips and are insufficient for evaluating long-term reasoning. I believe Long-RVOS will be a valuable and widely used resource for the community.
> > The proposed baseline, ReferMo, is simple yet competitive, and the additional experiments in the rebuttal clarify my remaining doubts.

---

> > > ### Author Response · Authors · 2025-08-04
> > >
> > > Thank you very much for your recognition and positive feedback. We sincerely appreciate your valuable suggestions throughout the review process. We will incorporate all the discussions and analyses from the rebuttal into our final version.

---

### Official Review · Reviewer_LPTM · 2025-06-27

**Clarity:** 3
**Significance:** 3
**Originality:** 3
**Rating:** 5
**Confidence:** 5

**Summary:**

This paper introduces Long-RVOS, a new benchmark dataset for referring video object segmentation (RVOS) in long-duration videos. Unlike existing datasets that focus on short video clips, Long-RVOS includes videos with an average duration exceeding 60 seconds, enabling the evaluation of long-term visual grounding. The dataset emphasizes static attributes, motion patterns, and spatio-temporal relationships. Two new evaluation metrics are proposed to assess temporal and spatio-temporal consistency. Furthermore, six state-of-the-art methods are benchmarked on Long-RVOS, and a baseline model, ReferMo, is proposed that incorporates motion cues to improve long-term understanding.

**Questions:**

1. Can the authors clarify how short clips are matched with full-length referring expressions? For example, if a referring sentence spans the entire video, how do you ensure that clip-level models are not misled by irrelevant temporal segments?
2. Why do SAM2-based methods underperform in Table 2?
3. How robust and time-efficient is the SAM2-based mask annotation process? Including the average annotation time and inter-annotator agreement (if any) would help validate the dataset’s quality.
4. It is encouraged to extend the dataset with reasoning-based referring expressions in the future.

**Ethical Concerns:**

["NO or VERY MINOR ethics concerns only", "Major Concern: Improper research involving human subjects"]

**Limitations:**

yes

**Quality:**

3

**Strengths And Weaknesses:**

Strength:
1. The paper is well-written and easy to follow. The motivation is clear. Compared with previous datasets, the proposed Long-RVOS includes long videos with object referring expressions, which is valuable for the community.
2. The dataset is built using manually written referring expressions and manually verified masks, facilitated by an adapted SAM2-based annotation tool. Figure 4 presents a word cloud that shows the diversity and frequency of descriptive attributes.
3. Six state-of-the-art methods are evaluated on Long-RVOS, and the paper includes an ablation study using the ReferMo baseline to demonstrate the challenge of long-term RVOS.


Weakness:
1. Compared to other benchmarks like ReasonVOS[1] and ReVOS[2], Long-RVOS lacks reasoning-based referring expressions involving spatio-temporal logic or contextual reasoning (e.g., object relationships, causality). While the dataset captures dynamic/static object descriptions, it does not fully reflect real-world reasoning demands. This limits its potential to evaluate higher-level model understanding.
2. As illustrated in Figure 5, the referring expression (e.g., “A student… boxing moves.”) spans the entire video, but clip-level retrieval requires aligning short video segments with appropriate referring expressions. The methodology lacks a clear explanation of how irrelevant text segments are filtered or avoided during short clip retrieval, which could introduce noise in evaluation.

[1] One Token to Seg Them All: Language Instructed Reasoning Segmentation in Videos. 2024, NeurIPS.
[2] VISA: Reasoning Video Object Segmentation via Large Language Models. 2024, ECCV.

---

> ### Author Rebuttal · Authors · 2025-07-30
>
> Thanks for your thoughtful and encouraging comments. We provide our repsonses as follows.
>
> > **Q1. Encourage extending the dataset with reasoning referring expressions.**
>
> Thanks for your suggestions. We fully agree that reasoning-based RVOS is a promising direction, as discussed in our Limitation section. In this work, we chose to begin with description-based RVOS because it is commonly used in current video applications and this task remains far from being solved. We will follow your suggestions to further extend Long-RVOS to reasoning scenarios in future versions.
>
> > **Q2. How are short clips matched with full-length referring expressions?**
>
> This is handled implicitly through the attention mechanism. Specifically, our model utilizes cross-attention to aggregate the vision and text features. During this process, text tokens irrelevant to the clip's content naturally receive lower attention weights, thus preventing the model from being significantly misled by the irrelevant text segments. Take Figure 5 as an example, the normalized attention weights in the 4-th encoder layer for "*boxing moves*" rise from 0.04 to 0.26 when this action occurs.
>
> > **Q3. Why do SAM2-based methods underperform on Long-RVOS?**
>
> While the recent SAM2-based methods excel in tracking and segmentation, they are typically limited in vision-language understanding, as discussed in Lines 298-307. In long videos with numerous distractors and events, it becomes challenging to accurately identify the referred target. Once the target object is misidentified, SAM2 will propagate the error throughout the entire video sequence, thus significantly degrading the performance. In contrast, our ReferMo identifies targets at keyframes as anchors and applies SAM2 within individual clips, effectively mitigating the potential error propagation and enhancing robustness in long-video scenarios.
>
> > **Q4. Average annotation time and inter-annotator agreement.**
>
> We used a rigorous multi-round workflow with separate teams for annotation and validation. The annotation time varies by case complexity: normal cases take **3-5 minutes** per object, while hard cases take **8-12 minutes**. The validation team consists of four experts. If an annotation is rejected by at least one validation expert, it will be returned back to the annotation team for re-annotation. Thank you for pointing it out. We will include these details in the final version.

---

> > ### Comment · Reviewer_LPTM · 2025-08-06
> > **Post-Rebuttal Comments**
> >
> > I would like to thank the authors for their thorough and thoughtful response. The clarifications are appreciated, and I have no further questions. I will retain my original score.

---

> ### Comment · Area_Chair_BqhG · 2025-08-05
> **Update after rebuttal**
>
> Dear Reviewer,
>
> The authors’ rebuttal has been posted. Please check the author's feedback, evaluate how it addresses the concerns you raised, and post any follow-up questions to engage the authors in a discussion. Please do this ASAP.
>
> Thanks.
>
> AC

---

> ### Author Response · Authors · 2025-08-06
>
> Thank you very much for your recognition and positive feedback! We truly appreciate your valuable input throughout the review process. We will incorporate all the discussions and suggestions from the rebuttal into our final version.

---

### Official Review · Reviewer_t4BS · 2025-07-02

**Clarity:** 3
**Significance:** 3
**Originality:** 3
**Rating:** 4
**Confidence:** 4

**Summary:**

This paper introduces Long-RVOS, a large-scale benchmark dataset for long-term referring video object segmentation containing 2,193 videos with an average duration of 60.3 seconds, significantly longer than existing datasets like MeViS (13.2s average). The dataset encompasses diverse challenging scenarios including occlusion and re-identification. The authors propose two new evaluation metrics specifically designed for long-duration videos and present ReferMo, a baseline method that achieves strong performance on the proposed benchmark using both conventional and newly introduced metrics.

**Questions:**

Could the authors explain why ReferMo shows such a large performance improvement (+2.8) over ReferDINO in the Static category in Table 2?

**Ethical Concerns:**

["NO or VERY MINOR ethics concerns only"]

**Final Justification:**

This paper presents a solid contribution, and I believe the dataset has the potential to positively impact the community. I support borderline acceptance.

**Limitations:**

The authors addressed limitations in appendix.

**Paper Formatting Concerns:**

No major formatting issues.

**Quality:**

3

**Strengths And Weaknesses:**

**Strength:**
- The paper addresses a highly important and practical problem in the field. Long-term referring video object segmentation is crucial for real-world applications but has been underexplored due to the lack of appropriate benchmarks.
- The motivation and storyline are clear and well-articulated. Despite covering substantial content, the paper is well-organized and easy to follow. The effective use of graphs and visualizations significantly enhances understanding of the proposed dataset and methodology.
- The dataset has sufficient volume (over 2K), and the average duration of 60 seconds presents appropriately challenging scenarios for current state-of-the-art models. The categorization of object sequences into dynamic, hybrid, and static types provides valuable analytical framework for understanding model performance across different tracking scenarios.
- The baseline method ReferMo has valid design and provides a reasonable starting point for future research on this benchmark.

**Weakness:**
The primary concern is the annotation quality of the dataset. According to the paper, mask annotations were generated using SAM2. Upon reviewing a few samples in the supplementary videos, I observed annotation issues, particularly in challenging occlusion scenarios. For instance, in video 9cc6ddb584_2.mp4, annotations are missing at critical frames (e.g., 11s, 39s). Video 4dee73ec95_1.mp4 shows representative cases of inconsistent mask quality in the 2-3s and 35-36s segments. Other samples exhibit the same issues. Annotation errors occur particularly in challenging scenarios involving occlusion, which ironically are the very situations this long-term benchmark aims to address. While RVOS tasks emphasize language-masklet alignment, video object segmentation fundamentally serves editing applications where IoU-based evaluation makes ground truth mask quality critically important. Although SAM2 also provided a semi-automatic annotation pipeline, their SA-V compensates for potential quality issues through massive scale (50K+ videos). Given Long-RVOS's scale, the annotation quality issues become more problematic and may limit the dataset's reliability for rigorous evaluation of long-term tracking methods. Overall, while the annotation quality is reasonable, it is not sufficiently robust for a definitive benchmark in this challenging domain.

---

> ### Author Rebuttal · Authors · 2025-07-30
>
> Thank you for your constructive feedback and appreciation of our work. We provide our repsonses as follows.
>
> > **Q1. Mask annotation quality.**
>
> Thanks for your valuable feedback. We fully agree that ensuring high-quality masks is important, and we've made many efforts to achieve this goal. As detailed in Lines 161-171, we developed an interactive mask annotation tool and conducted an iterative check-correction workflow with human annotators to minimize errors. Despite these efforts, given the significantly longer video duration in Long-RVOS (**60.3s** vs. **13.8s** in SA-V), it remains very challenging to ensure perfect masks across every frame. We respectfully believe that brief imperfect masks spanning **2-3** frames within a 60-second video (with **300+** frames) typically do not have a significant impact on training and evaluation. We will follow your suggestions to further refine the annotations and scale up the dataset in future versions.
>
> > **Q2. Why ReferMo outperforms ReferDINO in the Static category.**
>
> ReferDINO often fails on object-invisible frames because it tends to identify and segment the objects most relevant to the text in every frame. Instead, our ReferMo decouples the task into object identification in keyframes and mask propagation within clips via a pretrained tracker, leading to better robustness in long-term scenarios. Please see Figure 7 in Appendix for their qualitative comparison.

---

> > ### Comment · Reviewer_t4BS · 2025-08-05
> > **Post-Rebuttal Comments**
> >
> > Thank you to the authors for the rebuttal. While I acknowledge that the number of error frames is small relative to the total video length, I would like to reiterate that as models become more advanced, even minor inaccuracies in the ground truth can have a non-negligible impact. I appreciate the authors’ commitment to improving annotation quality in future releases. Given this, and considering the overall contribution of the paper, I will maintain my borderline accept recommendation.

---

> > > ### Author Response · Authors · 2025-08-05
> > >
> > > Thank you very much for your positive feedback and recognition of our work! We will follow your suggestions and continue to refine the mask quality in future versions.

---

> ### Comment · Area_Chair_BqhG · 2025-08-05
> **Update after rebuttal**
>
> Dear Reviewer,
>
> The authors’ rebuttal has been posted. Please check the author's feedback, evaluate how it addresses the concerns you raised, and post any follow-up questions to engage the authors in a discussion. Please do this ASAP.
>
> Thanks.
>
> AC

---

### Comment · Area_Chair_BqhG · 2025-08-01
**Authors' rebuttal posted and discussion**

Dear Reviewers,

Thank you for your efforts in reviewing this paper. The authors' rebuttal has been posted. This paper received diverse initial ratings. Please read the rebuttal materials and comments from other reviewers to justify if your concerns have been resolved and update your final rating with justifications.

---

### Decision · Program_Chairs · 2025-09-17

**Decision:**

Reject

**Comment:**

This paper studies an important problem of long-term referring video object segmentation, which has not been explored before. The authors propose a novel benchmark and a simple but effective baseline dedicated to this task. All reviewers acknowledge the significant contributions of the dataset presented in this work and give positive ratings. The AC agrees with the reviewers that the contribution of the proposed dataset is valuable, timely and crucial to facilitate development and related application of long-term referring video object segmentation. The baseline further demonstrates promising results.

However, since this paper's main contribution lies in building Long-RVOS and assessing existing methods, it is more appropriate for the Datasets & Benchmarks. Although a baseline is introduced, it primarily serves as a feasible solution with promising results for the proposed dataset rather than a substantive methodological advance. Therefore, according to NeurIPS policy, this work cannot be accepted in the research track (sorry for that) despite its potential impact on the community.